# School Choice at a Cost? Academic Achievement, School Satisfaction and Psychological Complaints among Students in Disadvantaged Areas of Stockholm

**DOI:** 10.3390/ijerph16111912

**Published:** 2019-05-30

**Authors:** Maria Granvik Saminathen, Sara B. Låftman, Bitte Modin

**Affiliations:** Centre for Health Equity Studies (CHESS), Department of Public Health Sciences, Stockholm University, SE-10691 Stockholm, Sweden; sara.brolin.laftman@su.se (S.B.L.); bitte.modin@su.se (B.M.)

**Keywords:** school choice, segregation, inequality, adolescents, well-being, school ethos

## Abstract

School choice allows students from more disadvantaged district areas in metropolitan Swedish cities to commute to more prestigious schools outside of their residential area. This study examined how such students fare compared to their peers who attend more deprived schools in their own district area. Multilevel analysis was applied, estimating 2-level random intercept linear regression models based on cross-sectional survey data collected among ninth grade students in 2014 and 2016 (*n* = 2105). Analyses showed that students living in relatively disadvantaged district areas of Stockholm who chose to attend more prestigious schools outside of their residential area performed better academically compared to students who opted to remain at more deprived schools in their catchment area, an association that was partly mediated by school quality in terms of teacher-rated school ethos. Yet, commuting students reported lower school satisfaction and more psychological complaints than students who stayed behind, even when taking academic achievement and school ethos into account. The association with psychological complaints was partly mediated by school satisfaction. Thus, the academic gain associated with having chosen to commute from a disadvantaged area to a more prestigious school does not appear to translate into higher school satisfaction and better psychological well-being.

## 1. Introduction

Sweden has become a rather unique case of universal school choice [1]. Unlike other contexts that have predominantly introduced small-scale voucher programmes, Sweden is one of few countries that provides all families with educational vouchers that are valid for both municipal and privately operated ‘independent’ schools [2,3]. Based on the idea that all families should have the opportunity to choose a school for their children [4], this policy was introduced as part of a series of educational reforms in the 1990s that intended to enhance educational quality, equity and efficiency [5,6]. The right to choose has since become an established programme [7], projecting that parents will select better schools for their children when provided with options. Especially in metropolitan cities characterised by deep residential segregation [8], the universal school voucher hence offers students in disadvantaged district areas the opportunity to apply for a placement in more privileged schools located in middle- or upper-class neighbourhoods. Admission criteria to comprehensive schools in Sweden are based on proximity and queueing rather than on student performance, thus enabling families to make such school choices regardless of the student’s academic success.

While parents and students tend to prioritise academic attainment during the final years of comprehensive school, this phase is also critical for adolescents’ psychological well-being. For adolescents and their psychosocial development, school presents a central environment, both as an educational and social arena [9,10,11,12]. Similar to other countries, Sweden has seen an increase in multiple health complaints among children and adolescents in recent decades [13], a trend that appears to apply to all sociodemographic groups [14]. Correspondingly, no significant differences in average psychological complaints was observed between schools with different sociodemographic profiles in a previous study of ninth grade students in Stockholm, although such schools differed substantially with regards to average academic performance [15]. Yet, it is nonetheless conceivable that there are discrepancies in psychological health between different groups of students within schools. For instance, there is evidence that children with immigrant background enjoy better self-reported mental health than children of native parents, despite commonly encountered obstacles such as discrimination and socioeconomic deprivation [16,17]. Such differences may depend on particular social contexts; both related to the individual family setting as well as shared spaces such as schools [11,13,17,18]. Concerning the school context, Hjern et al. [19] found that students born in Africa or Asia had a higher risk of being bullied or experiencing poor well-being in schools with few other migrant students. Such associations indicate that the sociodemographic composition of a school can play a role for student psychological well-being [20], which could be consequential for students living in more disadvantaged district areas of Stockholm who opt to commute to a socioeconomically more privileged school outside of their district area.

The following study focuses on how grade 9 students from relatively disadvantaged areas of Stockholm fare when they choose to attend more prestigious schools outside of their own school catchment area, concerning both their academic performance as well as their enjoyment of school and psychological well-being. 

### 1.1. Residential Segregation and School Choice in Metropolitan Sweden

The metropolitan Stockholm municipality is characterised by residential segregation based on socioeconomic status and immigrant background [21,22]. The so called ‘disadvantaged district areas’ of Swedish cities like Stockholm have experienced a development similar to the phenomenon of ‘white flight’ in the US context, with middle-class families and native Swedes gradually leaving for less troubled neighbourhoods [23,24]. Consequently, such areas suffer from rising marginalisation and low status [23], with a negative spillover effect on the schools. The predominantly lower-income and minority schools in such areas face a myriad of challenges, and thus they often struggle to maintain a conducive learning environment [25].

Inevitably, families with lower socioeconomic status have fewer options regarding their place of residence, and are thus restricted from relocating to better neighbourhoods in order to escape ‘failing’ schools [26]. On that account, school choice provides relatively disadvantaged children with better educational opportunities [27,28,29,30], as students are no longer confined to the schools in their respective catchment area [31,32]. However, Sweden’s school market has been criticised for reinforcing residential segregation patterns due to choice on unequal terms, as more privileged families are presently more likely to make an active and informed school choice, often to evade an undesirable student body composition. In addition, such families tend to reside in the areas where the most prestigious schools are located, which affords them priority to these schools due to the proximity system [33,34,35]. At the student-level, school choice may nonetheless have the potential to improve opportunities for the most socially disadvantaged students, since it allows their families to apply for a placement in more prestigious schools outside of their socially disadvantaged school catchment area at any point between grades zero and nine. 

In segregated cities like Stockholm, geographical location has become symbolic for social class and ethnicity [8], thus defining the social status of an area. Correspondingly, the sociodemographic student composition of a school affects its reputation and thus its desirability on the school market [36]. This is particularly true for schools in marginalised areas that tend to be the ‘losers’ of the social school hierarchy [37], partly due to ‘cream-skimming’ of more motivated and/or affluent students to higher status schools. While a school’s mean academic achievement is strongly correlated with the student body composition, it is mainly the sociodemographic profile that determines a school’s reputation and thus its symbolic value on the school market [38,39,40,41]. Accordingly, beyond geographical proximity and pedagogical practices, the symbolic capital of schools has become fundamental for parents navigating the school voucher market [8,42,43].

In Stockholm, such more privileged schools tend to be of better quality than schools with a more disadvantaged student body, for instance presenting with a more advantageous school ethos [15,44]. The concept of school ethos is part of a more comprehensive theory of school effectiveness [10] and refers to the beliefs, values and norms that shape the way that teachers and students relate, interact, and behave towards each other at the school [45,46]. Schools with a more sociodemographically advantaged student composition may be better equipped to build a strong school ethos, which in turn has been shown to positively affect student outcomes [10,47,48]. Therefore, by applying to more prestigious schools in more privileged residential areas, families in lower SES families are anticipating to escape the social stigma of their location and to provide their children with a more conducive learning environment, as well as more privileged peers and networks [42,49,50].

### 1.2. The Implications of School Choice for Socioeconomically Disadvantaged Students

The students in disadvantaged areas who decide to commute to more prestigious schools tend to be self-selected according to specific patterns [30]. Not only do they tend to be relatively high performing and motivated, but students with a native Swedish background, those with employed parents and/or parents with a relatively high education have also been shown to be more likely to transfer to schools outside of their more disadvantaged school district [51,52]. Nonetheless, attending a higher-SES school can provide access to a learning environment that is more favourable, with more structure and order, higher expectations from teachers and more academically driven classmates [44,53,54,55], which may enable students to thrive academically, regardless of their family background and previous performance. At the same time, this choice entails a longer commute to school, a separation from peers in one’s own neighbourhood, and contact with an unfamiliar school and peer culture [50]. Consequently, for such students, prospective gains in academic performance may not necessarily be indicative of equally positive non-cognitive outcomes, such as students’ enjoyment of school and their psychological well-being [56].

Thus, even if students who commute to more prestigious schools manage to perform better than their peers who attend a more deprived neighbourhood school [49,57,58], it is not inevitable that non-cognitive outcomes follow the same pattern. While it is generally projected that higher academic achievement is associated with better psychological well-being [13,59,60], the potential academic benefits of attending a more high-status school is not necessarily accompanied by affirmative feelings towards school and better psychological well-being. Openakker and Van Damme [61] have suggested that school characteristics act differently on well-being than on academic achievement. Their study in the Belgian context showed that classes effective in enhancing student achievement were not always simultaneously successful in cultivating student well-being. In England, Gibbons and Silva [62] revealed that students’ self-reported happiness and satisfaction with their learning environment was not higher in schools with higher average test scores. Similarly, in a US study, Crosnoe [63] found that low-income students experienced more psychosocial problems when attending schools with a majority of students from middle- or high-income families. Ackert [64] has written about the ‘segregation paradox’, showing that American minority background students who attended more affluent schools with high proportions of majority students were more engaged in coursework, but less likely to report that they like school. Bernburg et al. [65] showed that in Iceland, the effects of household economic deprivation on adolescent outcomes were stronger in schools were economic deprivation was rare. Such findings would indicate that students enjoy school more when their classmates have a more similar family background, and that attending higher-SES schools may have adverse effects on disadvantaged students’ psychological well-being. 

Yet, the effects of the academic environment and students’ own school performance may also be essential to take into account. Students tend to perform better when attending higher-achieving schools, regardless of their own background [33,58,66], and students with better school marks tend to have lower levels of psychosomatic symptoms than those with poor academic performance [11,13]. Further, schools also differ in terms of shared and practiced beliefs, values and norms. Schools with a more privileged student body composition tend to have higher teacher ratings of school ethos, on average, something which has been shown to be negatively associated with students’ psychological health complaints via a path going through their poorer academic achievement [67]. 

However, for relatively low-achieving students or adolescents without highly educated parents, social comparison mechanisms may come into play when attending schools with a majority of students from higher-SES families and/or higher average achievement, as these students are less likely to reach the top of their class in high-achieving schools [13,68]. Such experiences may have negative repercussions for their self-esteem and self-concept [63,69,70]. Further, academic competition may render them more stressed by schoolwork than in schools with lower average performance, particularly during the last year of comprehensive school. Accordingly, Swedish students who reported that their teacher assessed their performance as average or below average and those who felt stressed by schoolwork experienced higher levels of psychosomatic symptoms than other students [13,71]. 

In addition to such mechanisms, commuting commuting students may also struggle more with peer relationships in a more socioeconomically advantaged school setting. It is conceivable that students residing in the more sociodemographically disadvantaged and symbolically denounced areas of Stockholm find it more challenging to belong socially when attending higher-status schools than their peers who are enrolled at a school in their own catchment area. Considering the importance of peer connections during adolescence [12,72,73], such a lack of belonging can be expected to shape students’ satisfaction with school, with further implications for their well-being [36,74,75]. Such theories correspond to findings in a qualitative study by Bunar [36], who revealed that Stockholm students who deliberately stayed at deprived schools with poor reputations did so largely in order to preserve a feeling of safety and belonging, and in order to avoid stigmatisation due to their district of residence and family background. Thus, students from more disadvantaged areas attending higher status schools may experience the effects of socioeconomic deprivation more strongly than those attending more deprived schools [65,76], which may have consequences both for their academic achievement, as well as for their feelings towards school and their general well-being. 

### 1.3. Aim of the Study

In order to explore potential implications of school choice in terms of academic achievement, school satisfaction and psychological complaints, this study aims to examine how students living in the more disadvantaged district areas of Stockholm who have chosen to attend more prestigious schools outside of their residential area (hereafter movers) fare compared to their peers who are enrolled at more disadvantaged schools in their school catchment area (hereafter stayers). The research questions were as follows:(1)Do movers have higher academic achievement than stayers when taking student sociodemographic background characteristics into consideration?(2)Do movers report lower school satisfaction than stayers when adjusting for student sociodemographic background characteristics and academic achievement?(3)Do movers have more psychological complaints than stayers when adjusting for sociodemographic background characteristics and academic achievement?(4)Are any of the above associations mediated by school ethos?(5)Is any association with psychological complaints mediated by school satisfaction?

## 2. Materials and Methods

The study draws on combined cross-sectional teacher-student data from two separate data collections performed in 2014 and 2016. The student survey data was derived from the Stockholm School Survey (SSS) that Stockholm municipality conducts every two years among students in grade nine (aged 15–16 years) in all public and most independent schools in the municipality. Teachers’ ratings of school ethos were provided by the Stockholm Teacher Survey (STS) which was carried out by our research group via a web-based questionnaire and sent to all teachers (regardless of subject taught) working in Grades 7 to 9 in schools that had agreed to participate. The SSS was completed by 76% and 78% of the targeted Grade 9 students in 2014 and 2016, respectively. The corresponding figure for the STS is 54% (for both 2014 and 2016). School-level means of teacher-rated ethos were merged with student-level data to form a combined teacher-student data set covering information from a total of 10,757 students and 2,262 teachers across 169 middle school units. More information about the data material is provided elsewhere [77].

### 2.1. Study Sample

The analyses for this study were conducted on a sub-sample of the above described data material, with a restricted focus only on those students residing in the most sociodemographically disadvantaged district areas of Stockholm who either attended a more deprived school within their own residential area (*n* = 1325), or who commuted to a more prestigious school outside of their area (*n* = 780). 

First, we identified the district areas of Stockholm Municipality that could be considered the most sociodemographically disadvantaged. Official statistics from Stockholm Municipality [78] were reviewed to determine the most socially disadvantaged of the municipality’s 14 district areas. For this purpose, we examined the proportion of inhabitants with a tertiary education; the average income of the working population; as well as the proportion of inhabitants who were either born abroad or had two foreign-born parents. Based on these criteria, the following district areas were categorised as ‘most disadvantaged’: Rinkeby-Kista, Spånga-Tensta, Hässelby-Vällingby, Skärholmen and Farsta. While some of these areas are rather heterogeneous internally, they nonetheless present with the lowest proportions of residents with higher education, the lowest average incomes, and the highest proportions of inhabitants with a foreign background [78] ^i^. 

For the purposes of this study, the students residing in these district areas were grouped according to the location and segregation profile of the school that they attended. School segregation profiles were identified through latent class analysis of official data and consists of four distinct clusters of schools representing ‘privileged’, ‘typical’, ‘deprived’ and ‘deprived immigrant’ schools (for a more detailed description, see [15]). We distinguished between students attending schools within or outside of their district area of residency, and secondly, whether the school attended was classified as having more prestigious (‘privileged’ or ‘typical’) or deprived segregation profile (‘deprived’ or ‘deprived immigrant’). Based on this information, we created the variable school choice consisting of students who were classified as either ‘stayers’ or ‘movers’. Thus, students who attended more deprived schools in their own residential area were defined as ‘stayers’, whereas students who attended more prestigious schools outside of their respective area were defined as (upwardly mobile) ‘movers’. 

Besides excluding all students residing in more sociodemographically advantaged district areas from the original sample (*n* = 7447), we also removed students living in any of the selected five district areas who either attended more prestigious schools within their own area (*n* = 442), or who had chosen more deprived schools outside of their residential area (*n* = 638) ^ii^. This rendered a study sample of 2250 ninth-grade students distributed over 120 school units, covering 21% of the subjects in the original combined data. School aggregated information on school ethos for these 120 school units was based on a total of 1811 teacher ratings ^iii^. 

The final number of study subjects varied between the three outcome variables. For academic achievement and school satisfaction, complete information was available for 2105 and 1869 ninth-grade students distributed over 120 school units. The analyses for psychological complaints were based on 1655 students who were enrolled at 119 different school units. 

### 2.2. Measures

#### 2.2.1. Dependent Variables

Academic achievement was defined as the summation of the student’s self-reported marks in core subjects from the previous term, namely, Swedish, English, and mathematics. Response options were “No mark recorded” (student did not receive a mark in this subject) = 0; “Fail” = 0; “E” = 1; “D” = 2; “C” = 3; “B” = 4; and “A” = 5. The three items were added to form an index with the range 0–15. The measure showed good internal consistency (Cronbach’s alpha = 0.83). 

School satisfaction captured students’ self-reported sentiments about school. This index consists of four items: ‘I enjoy going to school’, ‘Schoolwork feels pointless’, ‘I look forward to going to my classes’, ‘Most of my teachers make learning interesting’. Students rated the accuracy of these respective statements on a four-point response scale ranging from describes very poorly (1), to describes very well (4). The school satisfaction index had a range of 4–16, with higher scores representing higher levels of satisfaction with school. The index showed reasonably good internal consistency (Cronbach’s alpha = 0.65). 

Psychological complaints were assessed by six items: ‘Do you feel sad and depressed without knowing why?’, ‘Do you ever feel frightened without knowing why?’, ‘Do you feel sluggish and uneasy?’, ‘How often do you feel it is really good to be alive?’ (reversely coded), ‘How much would you like to change yourself?’, ‘How often do you feel you’re not good enough?’. The first four items were rated on a five-point scale ranging from seldom (1) to very often (5). The item ‘How much would you like to change yourself?’ had five response alternatives, ranging from not at all (1) to very much (5), and ‘How often do you feel you’re not good enough?’ was rated on another five-point scale ranging from almost never (1) to very often (5). These six items formed an index with the range 6-30, with higher scores indicating more psychological complaints. The measure had good internal consistency (Cronbach’s alpha = 0.77).

#### 2.2.2. Independent Variables

School choice distinguished between ‘stayers’, referring to students who attend deprived schools in their own (disadvantaged) residential district area, and ‘movers’, referring to students who commute to more prestigious schools outside of their own (disadvantaged) residential area. 

#### 2.2.3. Control Variables

The analyses were adjusted for a number of student-level and school-level variables. Gender was captured by the question: “Are you a boy or a girl?”. The responses were coded into three categories: boys, girls, and information missing ^iv^. 

Parental education was measured by the question “Which is the highest education of your parents?” Four response options were provided separately for mothers and fathers: “Comprehensive school”, “Secondary school”, “University and university college”, and “I don’t know”. The responses were recoded into “No parent with university education or information missing”, “One parent with university education”, and “Two parents with university education”. This categorization has been used in previous studies of the same data material [15,67] ^v^.

In order to differentiate between students who were born in Sweden and those who have a migration background, we used the question “How long have you lived in Sweden?” with the response alternatives “All my life”, “10 years or more”, “5–9 years”, and “Less than 5 years”. This variable was recoded into “No migration background”, “Lived in Sweden 10 years or more”, and “Lived in Sweden 9 years or less”, in an attempt to (roughly) distinguish between students who arrived in Sweden before starting primary school and those who arrived in Sweden during primary school. Family structure was included as an additional control variable, differentiating between students who reported living with both parents and those who did not.

We further adjusted for the teachers’ ratings of the school’s ethos, assessed by 17 items in the STS. The measure of school ethos was designed to capture five sub-dimensions of the concept, namely (1) ‘Staff stability’, evaluating the level of sick-leave among teachers, staff turnover and the frequency of substitute teachers at the school, (2) ‘Teacher morale’, measuring whether the teachers have a strong work ethic, work with great enthusiasm, take pride in their school and feel confident as classroom leaders, (3) ‘Structure and order for dealing with unwanted behaviour’, through questions about the schools’ value system, whether the school actively works on issues such as violence, bullying and harassment among students, whether teachers feel confident about what they may and may not do if violent situations arise among students, and whether the rules for order and conduct are clear at the school, (4) ‘Teachers’ degree of student focus’, in terms of positive feedback to, and high expectations of the students, as well as whether the teachers take their time with students even if they want to discuss something other than schoolwork, and whether the students are treated with respect, and (5) ‘Academic atmosphere’, assessing whether the school provides a stimulating learning environment and whether the students’ motivation is a stimulating part of work. The response alternatives were “Strongly agree”, “Agree”, “Neither agree nor disagree”, “Disagree”, and “Strongly disagree”. The teachers’ ratings of the school’s overall ethos were aggregated to the school-level mean, and then z-transformed to have mean of 0 and a standard deviation of 1. The school-level means were subsequently merged with student-level data. The measure showed good internal consistency (Cronbach’s alpha = 0.92) and has been used in a previous study [46]. For the schools included in this study, the mean of the standardized variable of school ethos corresponded to −0.3, with a standard deviation of 1.2, indicating that the teacher-rated level of school ethos in the selected schools analysed in this study is generally lower than that of the full sample. This could be expected given the fact that many of the schools included in this study have a more deprived student body composition, and a previous study has shown that such schools tend to have lower levels of school ethos [15].

### 2.3. Ethics

Since the questionnaires in the Stockholm School Survey were filled in anonymously, data from the SSS are not subject to consideration for ethical approval, according to a decision made by the Regional Ethical Review Board of Stockholm (2010/241-31/5). Ethical permission for studies of the Stockholm Teacher Survey and the combined SST-STS data has been obtained by the Regional Ethical Review Board of Stockholm (2013/2188-31/5). 

### 2.4. Statistical Analysis

Considering that students were nested in schools and that one of the study variables was measured at the school level, the study used multilevel analysis. The statistical package was Stata/SE version 14.2 (StataCorp LLC, College Station, TX, USA). Analyses were conducted with two-level random intercept linear regression models of student outcomes [79], using the “xtmixed” command. Further, we tested for mediation with the “ml_mediation” command [80].

### 2.5. Analytical Strategy

In the multilevel regression analyses, we first estimated if there were any differences between stayers and movers in relation to academic achievement, school satisfaction, and psychological complaints. For each outcome, Model 1 presents b-coefficients adjusting for survey year, gender, parental education, migration background, family structure, and, in the case of school satisfaction and psychological complaints, academic achievement. Model 2 considered teacher-rated school ethos as an additional control variable. For the outcome psychological complaints, Model 3 further took school satisfaction into consideration. 

Finally, since a previous study [15] showed significant associations between the school’s segregation profile and its level of school ethos, we investigated whether school ethos mediated any of the associations between school choice groups and the three student outcomes. Further, since students’ satisfaction with school may contribute to their level of psychological well-being [81,82], we explored whether school satisfaction mediated any association between school choice group and psychological complaints. For all models, the Intra Class Correlation (ICC) was reported.

## 3. Results

Table 1 provides descriptive statistics regarding the distribution of student- and school-level variables between the two school choice groups, stayers (*n* = 1325) and movers (*n* = 780). Stayers attended schools with lower mean levels of teacher-rated school ethos (−0.47) than movers (−0.06), on average. Yet, the average level of ethos at schools attended by movers indicates that they did not principally attend the schools with the highest ratings of school ethos in Stockholm municipality. While a few movers were enrolled at one of the highest-rated schools, the majority of commuting students attended schools where the ethos had been rated as slightly above or below average. By contrast, more than half of stayers attended schools with ratings of school ethos that were markedly below the municipality’s average (data not shown). Further, among stayers, a lower proportion of students came from families where at least one parent had a higher education (36.7%) than among movers (54.2%). Further, considerably more stayers (20.2%) than movers reported having an immigrant background (7.0%). Finally, 66.7% of movers lived with both parents, compared 63.4% of stayers.

Table 2 shows results from random intercept models with students’ academic achievement, school satisfaction and psychological complaints, respectively, as the dependent variables, and school choice as the independent variable. First, we examined differences in academic achievement. Results showed that when adjusting for sociodemographic background characteristics, the average marks of movers was 1.05 (*p* < 0.001) units higher compared to stayers (Model 1). This lead was reduced to 0.93 (*p* < 0.001) units when further controlling for school ethos in Model 2. School ethos was found to have a significant mediation effect (b = 0.17, *p* = 0.032) on differences in academic achievement between stayers and movers, with an estimated mediation proportion corresponding to 15% of the total effect of school choice group on academic achievement.

For school satisfaction, Model 1 shows that movers’ satisfaction with school was lower than that of stayers when adjusting for sociodemographic background variables and academic achievement (b = −0.55, *p* = 0.001). Movers’ school satisfaction remained largely the same even when taking the school’s teacher-rated ethos (b = −0.58, *p* = 0.001) into consideration (Model 2). Accordingly, no mediation by school ethos was confirmed for the relationship between school choice and student school satisfaction. 

When estimating differences in psychological complaints, the results pointed to significant individual-level differences between stayers and movers, with movers’ estimated psychological complaints, on average, measuring 0.79 (*p* = 0.002) units higher than that of stayers, when controlling for student sociodemographic background and academic achievement (Model 1). This association dropped slightly after further adjustment for school ethos but remained statistically significant (b = 0.74, *p* = 0.004) (Model 2). Yet, when adding school satisfaction as a covariate in Model 3, the coefficient decreased markedly and the difference between stayers and movers became non-significant. We did not find any statistically significant mediation effect of school ethos for psychological complaints, but school satisfaction was found to be a strong mediating variable (b = 0.36, *p* = 0.001). Thus, school satisfaction explained 47% of the total effect of school choice group on levels of psychological complaints. 

## 4. Discussion

Previous research has suggested that school choice can be beneficial for families living in disadvantaged areas in residentially segregated cities like Stockholm, as they are free to opt out of neighbourhood comprehensive schools that may not live up to parents’ expectations [27,28,29,30,31,32]. While transferring out of a disadvantaged school district and into a more prestigious school could boost students’ school performance and future prospects [25], it is not evident that the implications for students’ enjoyment of school and their immediate psychological wellbeing are equally positive, due to various mechanisms [13,61,63,64,74]. 

This study explored how grade nine students who commute to a school outside of their own relatively more disadvantaged school district compare to their peers who have chosen to remain at a school in their own catchment area, examining both academic achievement, students’ satisfaction with school as well as their level of psychological complaints. Firstly, we observed that movers were more likely to have at least one highly educated parent and less likely to have an immigrant background, indicating an anticipated selection effect. Our finding regarding differences in academic achievement were in line with several previous studies [25,49,57,58]. Students whose families had made a conscious choice to enroll them at a higher-status school outside of their residential district area performed significantly better, on average, than their peers who have stayed behind at a school in their respective residential area, independent of their own family background. This finding is particularly pertinent considering that admission to comprehensive schools is not based on academic achievement. However, this choice of moving to a more sociodemographically privileged school outside of their home district area appears to come at a cost – the commuting students reported worse school satisfaction and more psychological complaints, on average, than those who attended schools with a more disadvantaged sociodemographic profile in their own school catchment area. Differences between movers and stayers persisted even when adjusting for individual family background characteristics and school ethos. 

Movers’ academic lead was expected, considering Swedish and international evidence regarding the benefits of attending higher-SES schools [25,44,49,57]. On the one hand, it is likely that a considerable proportion of families residing in disadvantaged areas who purposefully select a ‘better’ school for their children are distinctive with respect to certain observed and unobserved characteristics such as parental education, country of origin, parental involvement and support, as well as the student’s motivation and academic aptitude. Such indicators may thus be associated with the probability of making an active school choice as well as the school performance of commuting students in grade nine. 

Nonetheless, more sociodemographically privileged schools may be able to develop more favourable contextual features that in turn promote high academic achievement, benefiting all students at a school. At more socially deprived schools, teachers could also be assumed to adapt their instructional level and expectations to the average ability level of the students, further contributing to a less demanding and competitive academic culture [58]. Correspondingly, analyses showed that the school’s ethos acts as a mediator between school choice group and academic achievement. Although average ratings of ethos at schools attended by movers were below the average of all schools in the Stockholm School Survey, the average levels of school ethos at movers’ schools was nonetheless higher than at the lower-SES schools attended by stayers. This indicates that the relatively conducive school quality in terms of the way that teachers and students relate, interact, and behave towards each other in schools attended by movers partly explains differences in academic performance between movers and stayers. In addition to school contextual differences, sharing educational and social spaces with students from more educated backgrounds could potentially also contribute to increasing the performance of more disadvantaged students, by boosting their academic aspirations and motivation [55,83,84,85]. The findings of the present study imply that actively choosing a school with a more diverse or privileged student composition could indeed be indicative of higher academic performance for students living in disadvantaged areas. 

The results for school satisfaction and psychological complaints contradict notions relating to the well-established positive associations between school performance and psychological well-being among this particular group of students [13,59,67]. Instead, the conclusions are consistent with Ackert’s [64] ‘segregation paradox’ hypothesis. Accordingly, a more favourable sociodemographic composition may be associated with higher school performance, but due to the importance of peer connections during adolescence, school may present a more difficult social arena for movers than for stayers [12,64,73]. When transferring out of their school district, movers are separated from their neighbourhood peers and frequently confronted with classmates from higher-SES parts of Stockholm. Particularly movers who reside in the most stigmatised suburbs, those who have an immigrant background and/or whose parents are not highly educated may find it more challenging to find a sense of belonging at more prestigious schools [19,36,74]. Exposure to such a social context can be expected to play a role for students’ feelings towards school and in turn their psychological well-being [36,74,75]. 

Even when commuting students do not face discrimination from more affluent classmates as such, they are likely to encounter relative deprivation and more negative competition in schools with more high-achieving classmates from higher-SES backgrounds, with potential consequences for their school enjoyment and well-being [13,20,62,65,73]. Thus, one interpretation of our results could be attributed to social comparison mechanisms. Firstly, students tend to focus more on schoolwork in schools where many students have highly educated parents, creating a more competitive and possibly more stressful academic environment, particularly during the last year of comprehensive school. Moreover, commuting students from more socially disadvantaged backgrounds may fail to reach above-average levels of academic achievement on par with their more privileged schoolmates, with negative consequences for the individual’s self-concept [69,70]. Further, since more deprived schools tend to have been drained of the most motivated and academically talented students [23], the overall ambition level among students in such schools may be lower. Thus, in the short-term, stayers may feel less stressed and experience higher levels of well-being than their peers who are commuting to schools with a more privileged sociodemographic composition. Hence, although a highly ambitious academic environment could be stimulating and constructive for individual academic achievement, it could at the same time contribute to lower psychological well-being. This could also explain why school ethos did not mediate the association with school satisfaction and psychological complaints. Accordingly, such processes would undermine the value of the school choice structure for this group of students [56].

The analyses revealed that school satisfaction was a noteworthy mediator between school choice group and the level of psychological complaints, highlighting the importance of students’ school experiences for their psychological well-being. Thus, even if prestigious schools with a more privileged student composition appeal to many parents for educational purposes, students may have sound reasons for preferring to attend school in their more disadvantaged catchment area [36].

### Strengths and Limitations

This study identified and analysed two distinct groups of students in the Stockholm school choice landscape based on official as well as student- and teacher-reported data, and thus the findings were able to reveal different aspects of school choice for students living in disadvantaged areas. The teacher-level data added valuable school-contextual information that strengthened the results. 

Yet, due to the cross-sectional nature of the data, we cannot draw any conclusions about causality with support in the data. In addition, using self-reported student marks may compromise the validity of the measure academic achievement [86]. However, a previous study based on the Stockholm School Survey showed that the survey’s self-reported measures of marks in core subjects from the fall term in ninth grade did not differ substantially from corresponding official statistics for all Stockholm grade nine students [87]. The fact that “no mark recorded” and “fail” were combined into one category is acknowledged as an additional potential limitation of this variable.

Further, unobserved variables may explain some differences between movers and stayers [30]. For instance, restrictions in the data prevented us from explicitly identifying students with a foreign background, or compare students by country of origin. In addition, the measure of parental education was crude, partly due to high proportions of students with missing information, who were in the present study classified together with students with no parent with higher education. Moreover, although comprehensive schools cannot reject students based on their previous school performance, it is nonetheless plausible that many students in disadvantaged areas who actively seek out more prestigious schools could be more academically able and motivated than those who do not choose. Such selection effects are likely to have contributed to the observed gap in academic achievement between movers and stayers. 

Due to a lack of information regarding the time that a student has attended a particular school, we were unable to account for time effects on integration and school satisfaction among students, which may be relevant for movers in particular. Thus, longitudinal evaluations would be valuable. Finally, the distinctive school choice structure in Swedish metropolitan cities necessitates similar studies in other settings to substantiate the results. For instance, in school choice systems where only a limited number of low-income students are granted a voucher to transfer to a better school, outcomes may not correspond to the findings of this study, possibly due to selection effects. Further, in countries where educational resources are unequally distributed between school districts (depending on the tax base of the district), schools tend to differ more extensively in terms of quality than in Sweden, which could be expected to exacerbate potential differences between movers and stayers. However, considering the explanatory mechanisms that have been proposed by previous international research [19,20,36,64,69,70,73,74,80], it is likely that ‘movers’ would display similar patterns as in the present studies. 

## 5. Conclusions

The value of the school choice system is a matter of political ideology. Yet, regardless of distributional policy, it is evident that it is predominantly students residing in the most disadvantaged district areas who suffer the consequences of a segregated urban landscape. On the one hand, without the option to choose a school, all students in such areas would be assigned to their predominantly low-SES neighbourhood schools, limiting their access to the more advantageous social and cultural capital shaping higher-SES contexts [84], as well as potentially better-functioning schools [15,66,88]. Palpably, the school voucher policy has attempted to provide more equity by offering students in disadvantaged areas of Stockholm the opportunity to actively choose their school. On the other hand, one inadvertent consequence of school choice is the ‘cream-skimming’ of the most motivated and academically able students to ‘better’ schools in more privileged areas, further draining the most disadvantaged schools of human capital [2,23,36]. Moreover, as the findings of this study highlight, the potential relative advantage in academic achievement does not appear to translate into higher school satisfaction and better psychological well-being for students who commute to more prestigious schools. Indeed, as shown in previous studies [36,62,89], parental preferences regarding school quality and reputation are not always aligned with adolescents’ priorities of attending school together with more similar neighbourhood peers and avoiding the social stigma associated with one’s place of residence. This predicament faced by families in the more disadvantaged urban areas is one unfortunate consequence of incorporating a school voucher system in a highly residentially segregated city. 

Residential segregation and its causes are unlikely to be challenged in the short-term, but a revised method for school choice could promote a more diverse distribution of students between schools, with the potential to enhance educational equity and school quality [7,38,90]. In the meantime, schools with a more privileged student composition should consider developing their capacity to support commuting students from more disadvantaged areas in order to enable their successful integration into such schools. Future research could contribute to the current school choice literature by examining in more detail how sociodemographic school composition relates to contextual quality as well as disadvantaged students’ cognitive and non-cognitive outcomes, and how family features such as parental expectations and support interact with school choice.

## Figures and Tables

**Table 1 ijerph-16-01912-t001:** Distribution of school- and student-level variables for the study sample and by school choice group, (based on 2105 ninth-grade students distributed across 120 school units in five district areas in Stockholm municipality in 2014 and 2016).

Variables	All (*n* = 2105)			Stayers (*n* = 1325)			Movers (*n* = 780)		
	Mean	*SD*	Range	Mean	*SD*	Range	Mean	*SD*	Range
School-level									
School ethos									
Unstandardized	61.1	7.2	40.9–79.0	60.2	7.1	40.9−76.0	62.7	7.2	41.3–79.0
Standardized ^a^	−0.32	1.2	−3.7–2.62	−0.47	1.2	−3.7–2.1	-0.06	1.2	−3.6–2.62
Student-level									
Academic achievement	7.3	3.8	0–15	6.7	3.8	0–15	8.3	3.5	0–15
School satisfaction ^b^	11.1	2.5	4–16	11.3	2.5	4–16	10.9	2.5	4–16
Psychological complaints ^c^	14.0	5.1	6–30	13.8	5.0	6–30	14.3	5.3	6–30
	*N*	*%*		*N*	*%*		*N*	*%*	
Gender									
Boys	1001	47.5		617	46.6		384	49.2	
Girls	1046	49.7		673	50.8		373	47.8	
Information missing	58	2.8		35	2.6		23	3.0	
Parent(s) with university education									
None or information missing	1195	56.8		838	63.3		357	45.8	
One	466	22.1		277	20.9		189	24.2	
Two	444	21.1		210	15.8		234	30.0	
Migration background									
No	1782	84.7		1057	79.8		725	93.0	
In Sweden 10 years or more	180	8.5		143	10.8		37	4.7	
In Sweden 9 years or less	143	6.8		125	9.4		18	2.3	
Family structure (live with both parents)									
Yes	1,360	64.6		840	63.4		520	66.7	
No	745	35.4		485	36.6		260	33.3	

^a^ The statistical analyses are based on the standardized version of school ethos. Since the standardization of the measure was performed based on the entire sample of the original data (*n* = 10,757), the mean of the selected schools does not correspond to 0 and the standard deviation does not correspond to 1. Here, a mean of −0.32 in the study sample indicates that the school ethos in the selected schools analysed in this study is generally lower than that of the full sample. ^b^
*n*_all_ = 1869, *n*_stayers_ = 1166, *n*_movers_ =703, ^c^
*n*_all_ = 1655, *n*_stayers_ = 1015, *n*_movers_ = 640.

**Table 2 ijerph-16-01912-t002:** Results from two-level random intercept linear regression models (b coefficients). Student-reported academic achievement, school satisfaction and psychological complaints according to school choice group (based on all the available information of the ninth-grade students in the study sample taking part in the Stockholm School Survey in 2014 and 2016).

School Choice	Model 1	Model 2	Model 3
Academic achievement (*n* = 2105)			
Stayers (ref.)	0	0	
Movers	1.05 ***	0.93 ***	
ICC	0.046	0.035	
Mediating effect of school ethos ^a^		0.17 *	
School satisfaction (*n* = 1869)			
Stayers (ref.)	0	0	
Movers	−0.55 **	−0.58 **	
ICC	0.043	0.042	
Mediating effect of school ethos ^a^		n.s.	
Psychological complaints (*n* = 1655)			
Stayers (ref.)	0	0	0
Movers	0.79 **	0.74 **	0.42 (n.s.)
ICC	0.002	<0.001	<0.012
Mediating effect of school ethos ^a^			n.s.
Mediating effect of school satisfaction ^b^			0.36 ***

* Significant at the 5% level (*p* ≤ 0.05). ** Significant at the 1% level (*p* ≤ 0.01). *** Significant at the 0.1% level (*p* ≤ 0.001). ^a^ Test for mediation in the relationship between school choice and student outcome. ^b^ Test for mediation in the relationship between school choice and psychological complaints. Model 1: Adjusted for survey year, gender, parental education, migration background, and family structure (+ academic achievement in the analyses of school satisfaction and psychological complaints). Model 2: Model 1 + school ethos. Model 3: Model 1 + school satisfaction.

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
