# Peer review of "School Choice at a Cost? Academic Achievement, School Satisfaction and Psychological Complaints among Students in Disadvantaged Areas of Stockholm"

_ijerph, 2019, doi:10.3390/ijerph16111912_

Round 1

Reviewer 1 Report

Review of the manuscript “School choice at cost? Academic achievement, school satisfaction and psychological complaints among students in disadvantaged areas of Stockholm”

The authors conducted an interesting study. They compared a group of students from disadvantaged areas of Stockholm who  moved to more prestigious schools with students who stayed at schools in the disadvantaged areas. They showed higher academic achievement, but lower satisfaction with school and more psychological complaints in students who moved compared to students who stayed.  Associations were found to be partly mediated by school ethos and satisfaction with school.

The manuscript is well written. Presentation and interpretation of findings is good. Furthermore, strengths and limitations are well presented. I have some comments that might be considered and some suggestions that might further improve the manuscript:

Abstract:

Line 20: “these students”: It becomes clear later what is meant by “thee”. However, you may make it clear at hat point (students that moved to more prestigious schools) .

Introduction:

Line 29: “unique case of comprehensive school choice”. It is maybe because I am no native speaker of English. For me, it was not clear what is meant by “comprehensive school choice”.

The Introduction is well written. However, in my point of view, it is much too long. I suggest to shorten it and to focus on the essential things. It is difficult to say which parts could be shorten, maybe especially page 2 (first and last paragraph) and page 5, last paragraph.

Also, I suggest shorten and integrate paragraph 1.1. to the rest of the Introduction. It is interesting to read about the Swedish system. However, I believe that the same problems are encountered in other countries as well. The specificity of Sweden might be highlighted in a few sentences only.

I very much appreciated paragraphs 1.2. (1.2.1., 1.2.2), even if the last paragraph was too long. 

I think it should be 1.3. aim of the study (not 1.3.1)

Page 5, line 238: Please remove “second item” after research question 1.

I was missing the hypotheses.

Materials and Methods:

Page 6, line 248: “ninth and last grade”: What do you mean by this? Is the ninth grade also the last or did you include two different grades (ninth + last)?

Page 6, line 250: Can you add who performs the SSS?

Page 6, lines 270ff: can you please add the percentages of higher education, average income, and inhabitants with a foreign background (e.g., “[…] present with the lowest proportion of residents with higher education (between xx and yy%), the lowest average incomes (between xx and yy), and the highest […] background (between xx and yy%)”)?

I would add the information how many participants were movers and stayers already in the paragraph 2.1

Treatment of missing: I think it is not correct to code “no mark recorded” (which is like a missing, isn’t it?) as “0” and missing information on parental education as the lowest category of education. I suggest to exclude missings from analysis or (if there are too many missings) to replace them, e.g., to replace missing information on parental education on an appropriate median (e.g., median from that area). Did you really include parental background even if none of the parents provided an answer? I would at least exclude these cases.

I suggest to merge Tables 1 and 2. You can add three colums for each variable (total/overall; movers, stayers). The information given in both tables is similar. Furthermore, I would be interested in the mean scores of the dependent variables for stayers and movers.

Analytic strategy: It is not clear (at first sight) why you included academic achievement as control variable for dependent variables satisfaction and psychological complaints (and satisfaction as control variable for dependent variable psychological complaints). You may point out the role of these control variables in the Introduction.

Was the study approved by an ethics committee? How did participants consent to participate?

Results:

General remark: At which point are students allowed to move from one school (or area) to another? Can they change between grades? Did you assess how long students already stayed at their school (e.g., if the movers had changed before one year or many years before)? This might have an impact ion their integration and satisfaction. 

Table 3: I suggest to remove the lines  “School choice”.

Discussion:

First paragraph of the discussion: please add references.

Page 11, line 458: I suggest to add that the differences in academic performance were observed independently of social background (even if you state this at the end of the paragraph).

School ethos did mediate the association of school choice with academic achievement, but not the association with satisfaction and psychological complaints. Can you give an explanation for this?

I suggest to add a heading “limitations” in the Discussion (Page 12, last paragraph).

Page 13, lines 537-554: I would remove this from the conclusion as this is not what YOU have observed in the present study.

Author Response

Reviewer 1

Abstract:

Line 20: “these students”: It becomes clear later what is meant by “these”. However, you may make it clear at that point (students that moved to more prestigious schools).

We have added ‘commuting’ to clarify which students we are referring to here.

Introduction:

Line 29: “unique case of comprehensive school choice”. It is maybe because I am no native speaker of English. For me, it was not clear what is meant by “comprehensive school choice”.

Thank you for bringing this to our attention, this is indeed not very clear. We have changed it into ‘universal’, as this is a term that is commonly used when describing school choice systems that encompass all students (as opposed to a lottery system, or similar). I hope that this makes more sense.

The Introduction is well written. However, in my point of view, it is much too long. I suggest to shorten it and to focus on the essential things. It is difficult to say which parts could be shorten, maybe especially page 2 (first and last paragraph) and page 5, last paragraph.

Also, I suggest shorten and integrate paragraph 1.1. to the rest of the Introduction. It is interesting to read about the Swedish system. However, I believe that the same problems are encountered in other countries as well. The specificity of Sweden might be highlighted in a few sentences only.

I very much appreciated paragraphs 1.2. (1.2.1., 1.2.2), even if the last paragraph was too long. 

We have shortened the Introduction considerably, trying to highlight the aspects that are the most important for the present study. However, we have retained some of the background information about school choice in Sweden, as it is a rather unique system that is likely to be unknown to many readers. For instance, it is not self-evident that schools in Stockholm (in a country known for its emphasis on equality) differ so substantially in quality and student outcomes (partly due to the importance of the school’s ‘status’), despite the fact that schools in Sweden are supposed to be ‘equal’ with regards to resources (the most disadvantaged schools even receive additional funding to compensate for the student body’s socioeconomic background). Thus, we feel that some information on school choice in Sweden is significant for the understanding of the study and findings.

I think it should be 1.3. aim of the study (not 1.3.1)

This has been changed.

Page 5, line 238: Please remove “second item” after research question 1.

This has been removed.

I was missing the hypotheses.

We have considered and discussed the matter of including research questions versus hypotheses. When we conceived the study, we found that the nature of potential differences between movers and stayers was difficult to predict based on the previous literature (except for the differences in academic achievement). On the one hand, movers (are likely to) perform better and may thus also present with better non-cognitive outcomes based on the mutual relationship between performance and well-being, but at the same time these students find themselves in an unfamiliar environment away from their neighbourhood peers. As a result, we have concluded that research questions are more applicable than hypotheses in this study. Nonetheless, we have removed “higher” and “fewer” from research questions 2 and 3, respectively, to make the questions more focused.

Materials and Methods:

Page 6, line 248: “ninth and last grade”: What do you mean by this? Is the ninth grade also the last or did you include two different grades (ninth + last)?

We have simplified this by removing “last” (see also section 1.2., Further, academic competition may render them more stressed by schoolwork than in schools with lower average performance, particularly during the last year of comprehensive school.)

Page 6, line 250: Can you add who performs the SSS?

This information has been added.

Page 6, lines 270ff: can you please add the percentages of higher education, average income, and inhabitants with a foreign background (e.g., “[…] present with the lowest proportion of residents with higher education (between xx and yy%), the lowest average incomes (between xx and yy), and the highest […] background (between xx and yy%)”)?

We have added these statistics as an additional endnote in Appendix A (see i.).

I would add the information how many participants were movers and stayers already in the paragraph 2.1

This has been done.

Treatment of missing: I think it is not correct to code “no mark recorded” (which is like a missing, isn’t it?) as “0” and missing information on parental education as the lowest category of education. I suggest to exclude missings from analysis or (if there are too many missings) to replace them, e.g., to replace missing information on parental education on an appropriate median (e.g., median from that area). Did you really include parental background even if none of the parents provided an answer? I would at least exclude these cases.

With regards to the variable academic achievement, ‘no mark recorded’ is in fact a category in the SSS. These observations are thus distinctive from ‘missing’, as it indicates that the student has reported not having received a mark in this subject for some reason. We have now clarified this in parentheses (student did not receive a mark in this subject).

Thank you for your comment regarding the coding of the variable parental education. This variable is indeed problematic in the sense that up to 40% of students in our sample have not reported their parents’ education level.  However, we do not want to exclude these cases. In order to validate our findings, we have recoded the variable parental education to account for the missing observations. This new variable consists of four categories: “no parent with higher education”; “one parent with higher education”; “two parents with higher education”; and “information about parental education missing”. For transparency and clarity, we have presented the distribution of these four categories in Table 1. However, sensitivity analyses showed that the students in the category “information about parental education missing” performed largely the same as students without highly educated parents in relation to the three student outcomes in this study. Furthermore, including the variable with a separate category for missing observations in the multilevel regression analyses resulted in similar overall findings and thus would not have altered the conclusion of this study.

This has been explained in an endnote in the revised manuscript.

I suggest to merge Tables 1 and 2. You can add three columns for each variable (total/overall; movers, stayers). The information given in both tables is similar. Furthermore, I would be interested in the mean scores of the dependent variables for stayers and movers.

We agree that it makes sense to merge Tables 1 and 2. In this combined table, we have also included the mean scores of the dependent tables for stayers and movers.

Analytic strategy: It is not clear (at first sight) why you included academic achievement as control variable for dependent variables satisfaction and psychological complaints (and satisfaction as control variable for dependent variable psychological complaints). You may point out the role of these control variables in the Introduction.

We have tried to clarify this further in the Introduction. Considering the mutual relationship between academic achievement and non-cognitive outcomes, we wanted to hold academic achievement constant in the analyses of school satisfaction and psychological complaints to ensure that potential differences between movers and stayers were not explained by the higher average marks reported by movers. See section 1.2. (Yet, the effects of the academic environment and students’ own school performance…).

Was the study approved by an ethics committee? How did participants consent to participate?

Yes. We have included a section about ethical approval of the data used in the study.

Results:

General remark: At which point are students allowed to move from one school (or area) to another? Can they change between grades? Did you assess how long students already stayed at their school (e.g., if the movers had changed before one year or many years before)? This might have an impact on their integration and satisfaction. 

Unfortunately, the data do not contain information about when a student has changed schools (if at all). We only know which district area a student lives in and which school a student attends. Thus, we are able to know if a student has chosen to attend a school outside of their district area (and thus made an active school choice). We agree that it would have been very valuable to have information regarding school transfers. To clarify this, we have added the following sentence in Strengths and limitations:

Due to a lack of information regarding the time that a student has attended a particular school, we were unable to account for time effects on integration and school satisfaction among students, which may be relevant for movers in particular.

We have also highlighted that students can apply for a school transfer at any point during grades 0-9 (see section 1.1.)

Table 3: I suggest to remove the lines  “School choice”.

We have removed this line.

Discussion:

First paragraph of the discussion: please add references.

This is indeed needed; suitable references have been added as suggested.

Page 11, line 458: I suggest to add that the differences in academic performance were observed independently of social background (even if you state this at the end of the paragraph).

This has been clarified.

School ethos did mediate the association of school choice with academic achievement, but not the association with satisfaction and psychological complaints. Can you give an explanation for this?

Thank you for highlighting this. We have added the following sentences to the discussion to address this finding:

School ethos has been shown to be higher in socioeconomically privileged schools, and a strong school ethos is also known to be positively associated with student achievement [15], and inversely associated with psychological health problems [67]. The fact that the movers in the present study had higher academic achievement but lower school satisfaction and more psychological complaints than stayers could explain why school ethos mediated the association with academic achievement, but not the associations with school satisfaction and psychological complaints.

I suggest to add a heading “limitations” in the Discussion (Page 12, last paragraph).

We have added the heading 4.1 Strengths and limitations.

Page 13, lines 537-554: I would remove this from the conclusion as this is not what YOU have observed in the present study.

As advised, we have removed some of the content of the Conclusion and geared it more towards a conclusion of the study results. However, we have deliberately attempted to highlight the findings of this study in the context of the larger public discourse on school choice in Sweden, as we feel that this is highly relevant for the conclusion of our findings. In addition, even though this study focuses on ‘movers’, we also wanted to point out the consequences of school choice for ‘stayers’. School choice is clearly a complex matter, particularly as residential segregation is not likely to change in the near future. As a result, it is not obvious what action should/could be expected based on our findings. Thus, we found that such a conclusion was the most appropriate, while not taking an active (political) stand on the subject.

Reviewer 2 Report

The study entitled “School Choice at a Cost? Academic Achievement, School Satisfaction and Psychological Complaints among Students in Disadvantaged Areas of Stockholm” highlights the potential implications of school choice in terms of academic achievement, school satisfaction and psychological complaints. The study has been conducted well, the manuscript is well presented and the authors present the information in very detail. However, authors should be address some minor issues before the manuscript will be considered again for publication.

Material and methods

-          It is recommendable to include in this section that an Ethic Committee approved the study and talk about the inform consent (I suppose that it was like that)

-          More information about the psychometric properties of the measures used in the study is needed. For example, if applicable, it is necessary to report the reliability of each scale or instrument applied.

Discussion

-          Authors are aware that their research was carried out in Swedish and they say that the distinctive school choice structure in Swedish metropolitan cities necessitate similar studies in other contexts. But could be your results generalizable to other contexts? Which types of contexts? A statement about it is needed.

Author Response

Reviewer 2

Material and methods

-          It is recommendable to include in this section that an Ethic Committee approved the study and talk about the inform consent (I suppose that it was like that)

We have included a section about ethical approval of the data used in the study.

-          More information about the psychometric properties of the measures used in the study is needed. For example, if applicable, it is necessary to report the reliability of each scale or instrument applied.

We have incorporated Cronbach’s alpha for the three dependent variables to report the reliability of the indices (see 2.2.1.) as well as for the control variable school ethos (see 2.2.3.).

Discussion

-          Authors are aware that their research was carried out in Swedish and they say that the distinctive school choice structure in Swedish metropolitan cities necessitate similar studies in other contexts. But could be your results generalizable to other contexts? Which types of contexts? A statement about it is needed.

This is a good point - we have added a brief discussion about this matter at the end of the Discussion section (under Strengths and Limitations).

Reviewer 3 Report

Dear authors,

I consider this is an interesting manuscript about the cost of school choice in students of disadvantaged areas of Stockholm who chose to attend more prestigious schools outside of their residential area. In this manuscript not only academic achievement but also school satisfaction and students’ psychological complaints are analyzed. The inclusion of these three variables, and the relationships among these, is a positive aspect of this study. The present manuscript is adequately executed and provides interesting results.

Introduction is well organized and the different subsections are adequate. The aim of the research is clearly established.

Method. The research design is appropriate. Sufficient information about the sample, sociodemographic characteristics of their families, and school characteristics is provided. Perhaps, if the authors have these data, it could be also included information about the number of siblings. Also, it would be convenient to indicate the age mean of the final sample included in this study.

In the description of the instruments used to evaluate the variables of school satisfaction and psychological complaints should be included information about their psychometric properties (reliability of the scale, factor analysis), and how these scales were elaborated (for example, if they are based on other previous scales). Also, information about the reliability of the measure of school ethos should be provided.

Results are clearly described, and tables are adequately presented.

The conclusions are supported by the results, and interesting ideas are provided. In future research it could be convenient to include also variables related to family functioning, styles of socialization, and parental expectations about the education of their children. A more complete perspective could be achieved by analyzing some of these family variables.

Author Response

Reviewer 3

Method. The research design is appropriate. Sufficient information about the sample, sociodemographic characteristics of their families, and school characteristics is provided. Perhaps, if the authors have these data, it could be also included information about the number of siblings. Also, it would be convenient to indicate the age mean of the final sample included in this study.

Unfortunately, the Stockholm School Survey (SSS) does not provide information about the number of siblings. Further, we are unable to calculate the mean age of the sample, as the question about the student’s age in the SSS is limited to the categories ’15 years or younger’; ’16’; ‘17’; ‘18’; and ’19 years or older’. However, 60.4% of the final sample reported being 15 years or younger; and 37.5% indicated that they are 16 years old. Age 15-16 years corresponds with the common age for grade 9 (since children in Sweden start grade one the year that they turn 7 years).

In the description of the instruments used to evaluate the variables of school satisfaction and psychological complaints should be included information about their psychometric properties (reliability of the scale, factor analysis), and how these scales were elaborated (for example, if they are based on other previous scales). Also, information about the reliability of the measure of school ethos should be provided.

We have incorporated Cronbach’s alpha for the three dependent variables to report the reliability of the indices (see 2.2.1.) as well as for the control variable school ethos (see 2.2.3.).

Results are clearly described, and tables are adequately presented.

The conclusions are supported by the results, and interesting ideas are provided. In future research it could be convenient to include also variables related to family functioning, styles of socialization, and parental expectations about the education of their children. A more complete perspective could be achieved by analyzing some of these family variables.

Thank you for this input; we have added this suggestion to the final paragraph of the article.

Round 2

Reviewer 1 Report

The authors addressed each comment and the manuscript has been improved. The paper is interesting and easy to read. However, I would like to insist on one study limitation, which should at least be mentioned in the limitations section. This limitation concerns the categorization of “no mark recorded” (as the student did not receive a mark) and “Fail” into the same category and the categorization of “missing information on parental education” and “No parent with university education” into the same category. “No mark recorded” should not be equated to fail (even if it is no missing). Please exclude cases with no marks recorded, use another category or – at least – mention your strategy as a limitation.  For educational background, you mentioned that children with information missing did not differ from children of families with no university education. One could wonder why, in that case, you still decided to merge both categories… I suggest to explain directly in the text (not only in the Appendix) why both categories were merged. Otherwise, the differences between categories mentioned in text and in the Table are confusing.

Typing errors: page 6, line 288: teachers’; Page 11, line 465: achievement on par with; Page 12, line 520: One the one hand

Page 11, lines 477ff: I do not understand your explanation why school ethos mediated academic achievement but not school satisfaction and psychological complaints. I suggest to make your point clearer/to find another explanation or to remove this passage. 

Author Response

Reviewer 1

However, I would like to insist on one study limitation, which should at least be mentioned in the limitations section. This limitation concerns the categorization of “no mark recorded” (as the student did not receive a mark) and “Fail” into the same category and the categorization of “missing information on parental education” and “No parent with university education” into the same category. “No mark recorded” should not be equated to fail (even if it is no missing). Please exclude cases with no marks recorded, use another category or – at least – mention your strategy as a limitation.  

This should indeed be mentioned, thank you. We would prefer to keep the coding of the variable measuring academic achievement (partly as it is consistent with previous studies), but have now included the fact that “no mark recorded” and “fail” were combined into one single category as a limitation on p. 11.

For educational background, you mentioned that children with information missing did not differ from children of families with no university education. One could wonder why, in that case, you still decided to merge both categories… I suggest to explain directly in the text (not only in the Appendix) why both categories were merged. Otherwise, the differences between categories mentioned in text and in the Table are confusing.

Thank you for assisting us to make sure that this is as clear and transparent as possible. We have discussed this once again and agree that it may appear confusing. One reason why we would like to keep the original coding of the variable ‘parental education’ is consistency, as this study is linked with previous studies on the same data and similar topic. In order not to lose statistical power, these previous studies combined “no parent with higher education” and “no parental education reported” (after conducting sensitivity analyses). Hence, we have decided to present the variable with three categories in Table 1 (as in the first version of this manuscript), to briefly mention the matter in the text and to include the following explanation in endnote v in Appendix A:

In the study sample, the category including “no parent with higher education or information missing” consisted of 1,195 students, of whom 455 (21.6% of the study sample) had not reported that any parent had higher education, and 740 (35.2% of the study sample) had missing information. Sensitivity analyses showed that the students with no parent with higher education and students with missing information on parental education performed largely the same in relation to the three student outcomes in this study. Furthermore, including the variable with a separate category for missing observations in the analyses resulted in similar overall findings and would thus not have altered the conclusion of this study.

We have added a sentence acknowledging the limitations with our measure of parental education (pp. 11-12).

Typing errors: page 6, line 288: teachers’; Page 11, line 465: achievement on par with; Page 12, line 520: One the one hand

Thank you for pointing out these typing errors. However, we are not sure what you would like us to change with regards to “on par with”?

Page 11, lines 477ff: I do not understand your explanation why school ethos mediated academic achievement but not school satisfaction and psychological complaints. I suggest to make your point clearer/to find another explanation or to remove this passage. 

The explanation that we added was statistical rather than theoretical. Considering the direction of association between school choice category and school satisfaction and psychological complaints that the findings revealed (Model 2), it would not have made sense for school ethos to be a mediator. However, we did not know for sure what the association would be when we conceived the study. We have since removed this passage and instead written the following (p. 11):

Hence, although a highly ambitious academic environment could be stimulating and constructive for individual academic achievement, it could at the same time contribute to lower psychological well-being. This could also explain why school ethos did not mediate the association with school satisfaction and psychological complaints.
